# PAPPA Expression in Indeterminate Thyroid Nodules as Screening Test to Select Patients for Molecular Testing

**DOI:** 10.3390/ijms23094648

**Published:** 2022-04-22

**Authors:** Carlotta Marzocchi, Silvia Cantara, Alfonso Sagnella, Maria Grazia Castagna

**Affiliations:** 1Department of Medical, Surgical and Neurological Sciences, University of Siena, Viale Bracci 1, 53100 Siena, Italy; carlottamarzocchi@libero.it (C.M.); alfonso.sagnella@student.unisi.it (A.S.); mariagrazia.castagna@unisi.it (M.G.C.); 2Laboratory of Clinical and Traslational Research, AOU Siena, Viale Bracci 1, 53100 Siena, Italy

**Keywords:** PAPPA, FNAC, indeterminate thyroid nodules, thyroid cancer, molecular test

## Abstract

Pregnancy-associated plasma protein A (PAPPA) acts as an oncogene, and its expression is increased in multiple malignancies, including thyroid cancer. Molecular tests represent a useful tool in the management of indeterminate thyroid nodules; however, they are not conducted in all centers, and they contribute to increase the per-patient cost of nodule evaluation. In this study, we examined whether PAPPA expression could represent a promising new screening test in the management of indeterminate thyroid nodules. Toward this aim, PAPPA expression was evaluated in 107 fine needle aspiration cytologies (FNAC) belonging to Bethesda III–IV categories that had been sent to molecular biology to discriminate the nature of the nodules. We found that the PAPPA expression increased and showed an elevated sensitivity (97.14%) and negative predictive value (98%) in indeterminate cytological samples positive for mutations. The enhanced expression was not linked to a specific oncogene. Our findings demonstrated that assessing the PAPPA expression in indeterminate thyroid cytologies could represent a useful screening tool to select all patients that effectively need to be sent to molecular testing, thereby, leading to a potential cost reduction in the management of patients.

## 1. Introduction

Pregnancy-associated plasma protein A (PAPPA) is a metalloproteinase that interacts with insulin-like growth binding proteins (IGFBPs), especially IGFBP4, regulating proteolysis and bioavailability of insulin-like growth factor ligand 1 (IGF1) [1]. Circulating PAPPA represents a consolidated biomarker in combined first-trimester screening tests, allowing an early prenatal diagnosis of chromosomal abnormalities [2,3], preeclampsia [4] and intrauterine growth restriction risk [5].

In addition to pregnancy, PAPPA was found to be increased in several tumors, including thyroid cancer [6,7]. Studies have demonstrated that PAPPA acts as an oncogene by promoting cancer cells proliferation, migration and invasion [8]. In light of these data, PAPPA may be considered as an interesting potential target for the treatment of tumor progression [8]. We first demonstrated that PAPPA behaves as a promising diagnostic marker for differentiated thyroid cancer contributing to the pre-surgical classification of thyroid nodules according to the final histology [7].

Early diagnosis of thyroid cancer represents a priority as thyroid nodules are increasing in the general population in all countries [9]. The majority of thyroid nodules are benign hyperplastic nodules, and thyroid cancer is found in less than 10% [10]. Fine needle aspiration cytology (FNAC) represents the gold standard to discriminate benign from malignant nodules despite limitations due to inadequate or indeterminate samples [11]. 

Only about 10% to 30% and 25% to 40% of Bethesda categories III and IV (indeterminate cytologies), respectively are malignant at the final histology [12]. This means that a high proportion of patients with indeterminate FNAC will undergo unnecessary thyroid surgery. The discovery of genetic alterations in differentiated thyroid cancer prompted the search of somatic mutations in material obtained by FNAC, aimed to increase the diagnostic accuracy of traditional cytology and to help clinicians in the appropriate clinical decision [13]. 

To further increase the sensitivity and specificity over the years, other markers (i.e., microRNAs, immunocytochemistry and proteomics markers) have been proposed together with molecular test; however, none of these have provided adequate results or been introduced in clinical practice [14]. Molecular testing has actually evolved from single mutation evaluation to more broad genetic panels conducted using advanced technologies, such as next generation sequencing [15]. As a result, molecular testing must be performed in specialized laboratories and by qualified professionals; however, most centers lack facilities and the appropriate expertise.

In our study, we aimed to evaluate whether the expression of PAPPA in indeterminate FNAC could be used as a screening tool to select patients to be sent for advanced molecular diagnostic testing.

## 2. Results

### 2.1. Molecular Characterization of FNAC

A seven-gene panel of mutations was investigated as 107 FNAC samples. From the analysis, 35 out of 107 (32.7%) samples were found positive for one mutation, and 72 out of 107 (67.3%) were negative. RAS oncogenes were more frequently mutated (17/35, 48.6%) with 9/17 (52.9%) NRAS, 7/17 (41.2%) HRAS and 1/17 (5.9%) KRAS. BRAF mutation(s) was found in 13/35 cases (37.1%) with 12/13 (92.3%) BRAF V600E and 1/13 (7.7%) BRAF K601E. Four/35 (11.4%) cases harbored RET/PTC1 rearrangements, and only 1/35 cases (2.9%) showed PAX8/PPARγ rearrangement. No RET/PTC2 or RET/PTC3 rearrangements were found. The results of the molecular biology tests are summarized in Table 1.

### 2.2. PAPPA Expression Increased in Positive Mutations Cytological Samples

PAPPA mRNA expression was analyzed by qPCR in 107 cytological samples characterized for mutations. PAPPA was significantly higher (*p* < 0.0001) in cytological specimens positive for mutations (n = 35) compared to negative cases (n = 72) (Figure 1a). The group of positive mutations consists of 27/35 (77.1%) samples belonging to Bethesda categories III–IV and 8/35 (22.9%) to Bethesda categories V–VI. These latest categories were included for the comparison of all samples regarding suspicion for/malignancy. 

Cytological specimens negative for mutations were all Bethesda categories III–IV (Figure 1a). To verify if the Bethesda V–VI samples with positive mutations could influence statistical significance, the same analysis was conducted excluding them. Again, considering only samples belonging to Bethesda categories III–IV positive for mutations (n = 27), PAPPA levels were significantly increased (*p* < 0.0001) compared to Bethesda categories III–IV negative for mutations (Figure 1b). The extent of PAPPA expression was similar between Bethesda III–IV positive for mutations and Bethesda V–VI (n = 8) (*p* = 0.49) (Figure 1c). Finally, no correlations were found between PAPPA levels and specific gene alterations (*p* = 0.44) (Figure 1d).

### 2.3. PAPPA Expression Showed a High Sensitivity and Negative Predictive Value to Identify Nodules with Genetic Alterations

We evaluated whether PAPPA expression could be used as a potential biomarker to screen nodules with oncogene mutations respect to those with negative mutations. As shown in Figure 2a by the receiver operating characteristic (ROC) curve area of 90.1% (95% confidence interval: 84.5–95.6%; *p* < 0.0001), PAPPA mRNA levels displayed a high degree of diagnostic accuracy to distinguish cytologies that were positive for mutations from those negative for mutations. 

Our criterion for evaluating the optimal threshold of the test was to maximize the sensitivity and minimize false negative results in order to correctly select patients to be sent to molecular testing. The calculated Youden cut-off of 0.02732 showed a sensitivity of 97.14% and a specificity of 69.44% with an overall accuracy of 78.5%, a negative predictive value (NPV) of 98% and a positive predictive value of PPV of 60.7% (Figure 2b).

## 3. Discussion

The majority of thyroid nodules are benign, and approximately 10% of them are cancerous [10]. In this view, it is very important for the pre-surgical diagnosis to distinguish benign from malignant thyroid nodules in order to limit surgical treatment only to the malignant/suspicious ones. Currently, FNAC represents the “gold standard” for the differential diagnosis of thyroid nodules. In general, in expert hands, it is associated with good specificity and sensitivity [11]. However, this procedure has some limitations related to inadequate sampling (2–16%) or to the difficulty to discriminate follicular lesions (5–20%). Among the indeterminate samples, only a proportion are malignant at the final histology [12]. 

The discovery of genetic alterations specific for differentiated thyroid cancer may provide molecular markers to be searched for in the material obtained by FNAC, thus, increasing the diagnostic accuracy of traditional cytology. Several molecular tests have been introduced in the clinical routine, such as the Gene Expression Classifier (GEC), which recognizes benign lesions on the basis of an expression pattern of mRNA [16], a seven-gene mutational panel [17,18] and targeted next-generation sequencing (tNGS) [19,20,21]. 

Mutation panels aimed to identified malignancies must include at least BRAF, RAS point mutations and RET/PTC and PAX8/PPARγ rearrangements. “Homemade” methods, including PCR with final sequencing or commercial kits based on real time PCR (qPCR), are available. The use of tNGS with larger panels of genes is an alternative method capable to attain a NPV of 95% or more and a high PPV and sensitivity [13]. 

Although molecular tests represent a useful tool in the management of indeterminate nodules [13], they are not conducted in all centers and contribute to increasing the cost per-patient for nodule evaluation. In this optic, to optimize personalized medical care and management algorithms, PAPPA expression on FNAC could represent a promising, new, quick, cheap and easy screening tool for correctly selecting patients with indeterminate cytology in which molecular investigations are helpful (Figure 3). PAPPA mRNA levels are increased in positive for mutation cytologies, and, with a calculated Youden cut-off of 0.02732, PAPPA reached a sensitivity of 97.14% with a negative predictive value of 98%, thereby, strengthening the results of our previous work [7].

The study has a few limitations since histological confirmation was available only in a small subgroup of thyroid nodules. Indeed, histology was available in 8/27 Bethesda III–IV samples positive for mutations. We have to consider that we are a reference center for the management of indeterminate lesions, and we received samples for the genetic analysis from several endocrinology units widespread for Italy. Not all patients are submitted to surgery in our clinic. However, the correlation between histology and PAPPA expression was coherent as we already published in our previous work [7]. 

Furthermore, all patients followed in our center with indeterminate nodule(s) negative for mutations (n = 72) preferred a conservative approach, and, due to the lack of clinical evidence, these patients were not submitted to surgery. In the follow up period, nodules did not increase in diameter, and the ultrasound characteristics, in terms of the echogenicity, presence/absence of microcalcifications and margins, remained the same (data not shown). However, the study has several strengths, such as a standardized management in the same institution with detailed information regarding the long term follow-up of thyroid nodules not submitted to surgery. 

Finally, to our knowledge, this is the first study that evaluated the usefulness of a screening test in selecting indeterminate thyroid nodules worthy of molecular analysis. Our preliminary results seem to suggest that PAPPA expression can be performed on indeterminate FNAC before the molecular tests in order to identify patients at risk of having cancer and that will benefit from genetic analysis. The introduction of this test in the practice could reduce the costs for the management of indeterminate lesions. Nevertheless, our results need to be confirmed in a larger series of indeterminate thyroid nodules characterized for mutations.

## 4. Materials and Methods

### 4.1. Patients

This study retrospectively analyzed 107 consecutive thyroid nodules belonging to 104 patients who underwent to FNAC from 2015 to 2021 and of which the cytological material was still available in our bank. Our cohort of patients consist of 84 females (80.8%) and 20 males (19.2%). Thyroid cytopathologies were reported using the Bethesda System. We found that 79/107 (73.8%) were Bethesda III, 20/107 (18.7%) were Bethesda IV, and 8/107 (7.5%) were Bethesda V–VI. Only 15 patients underwent thyroid surgery. At the final histology, one nodule Bethesda III was found with adenoma, seven Bethesda IV were found to be malignant (one hurtle, one follicular thyroid cancer (FTC) and three papillary thyroid cancers (PTC), two were PTC follicular variant (PTCFV)), and all Bethesda V–VI were confirmed thyroid cancers (seven PTC and one PTCFV).

Informed consent was signed by each patient enrolled in the study, and the study was approved from our local Ethical Committee (Ethics Committee of Regione Toscana, Area Vasta Sud Est, AOUS. Protocol ID: 10167).

### 4.2. RNA and DNA Isolation from Cytological Material

Cytological material, achieved after the fine-needle aspiration biopsy procedure, was preserved in two aliquots of 150 µL/each of RNAprotect Cell Reagent (Qiagen, Hilden, Germany) before RNA and DNA extraction. RNA and DNA were isolated using an RNeasy mini kit (Qiagen, Hilden, Germany) and QIAamp DNA blood mini kit (Qiagen, Hilden, Germany), respectively, following the manufacturer’s instructions. The nucleic acid quantity and quality were assessed using NanoDrop One (Thermo Scientific, Waltham, MA, USA).

### 4.3. Oncogene Mutations Analysis

All samples were genotyped using a qPCR procedure. Specifically, RET/PTC1, RET/PTC2, RET/PTC3 and PAX8/PPARG rearrangements were evaluated using RNA with a EasyPGX^®^ ready THYROID Fusion kit (Diatech Pharmacogenetics, Jesi, Italy). Point mutations were searched on DNA at codons 12,13 (exon 2) and 61 (exon 3) of KRAS, NRAS and HRAS genes, and at codons 600/601 of BRAF gene using a EasyPGX^®^ ready THYROID kit (Diatech Pharmacogenetics, Jesi, Italy). Samples were run on the thermal cycler EasyPGX qPCR instrument 96 (Diatech Pharmacogenetics, Jesi, Italy), and the data were analyzed with the “EasyPGX Analysis Software” (Diatech Pharmacogenetics, Jesi, Italy).

### 4.4. PAPPA Evaluation by Quantitative Real Time PCR

Two-hundred nanograms of RNA for each sample were retrotranscribed with M-MuLV-RH First Stand cDNA Synthesis Kit (Experteam, Venice, Italy). The expression levels of PAPPA were evaluated with qPCR using FastStart Essential DNA Green Master Mix (Roche, Basilea, Switzerland) on the Rotor-Gene Q real time PCR (Qiagen, Hilden, Germany). Each sample was run in triplicate and normalized against actin (ACTB, Eurofins, Luxemburg, Luxemburg). The quantification of the PAPPA expression levels was determined using the 2-ΔCT method. The specific primers were PAPPA PF: 5′-TGAATCTGAGCAGCACATTG-3′ and PR: 5′-CATCGTCTTCCAAGCACTTC-3′; and ACTB PF: 5′-CACCAACTGGGACGACAT-3′ and PR: 5′-ACAGCCTGGATAGCAACG-3′.

### 4.5. Statistical Analysis

All statistical analyses were conducted using the software GraphPad Prism version 5. A value of *p* < 0.05 was considered to be statistically significant. For the qPCR analysis, the normality of the data was assessed using the Shapiro–Wilk test. Statistical differences in the PAPPA mRNA levels were verified using the two tailed Mann–Whitney U test to compare two groups and by the Kruskal–Wallis H test, followed by Dunn’s test, to compare two or more independent groups. The accuracy of the PAPPA mRNA expression as a diagnostic test was evaluated by the ROC curve and the AUC.

## Figures and Tables

**Figure 1 ijms-23-04648-f001:**
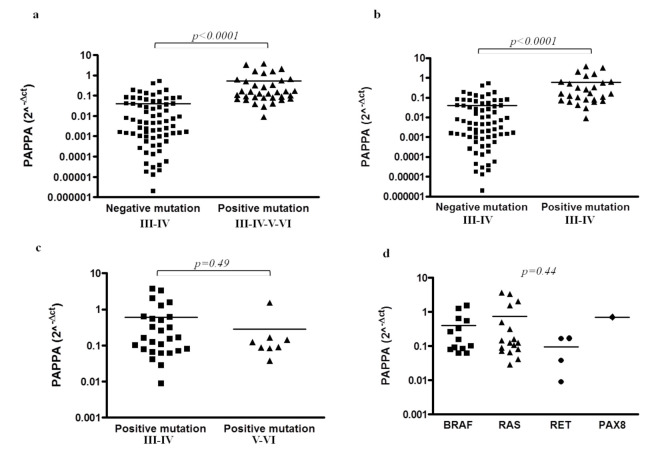
Pregnancy-associated plasma protein A (PAPPA) expression mRNA levels in cytological samples. (**a**) PAPPA expression in Bethesda III–IV with negative mutations (n = 72) compared to Bethesda III–IV-V–VI with positive mutations (n = 35) samples. *p* < 0.0001 by the Mann–Whitney U test; (**b**) PAPPA expression in Bethesda III–IV with negative mutations (n = 72) compared to Bethesda III–IV with positive mutations (n = 27) samples. *p* < 0.0001 by the Mann–Whitney U test; (**c**) Comparison of PAPPA expression in positive for mutations samples belonging to Bethesda categories III–IV (n = 27) and V–VI (n = 8). *p* = 0.49 by the Mann–Whitney U test; (**d**) PAPPA mRNA expression stratified according with the specific gene mutations in BRAF (n = 13), RAS (n = 17), RET (n = 4) and PAX8 (n = 1). *p =* 0.44 by the Kruskal–Wallis test.

**Figure 2 ijms-23-04648-f002:**
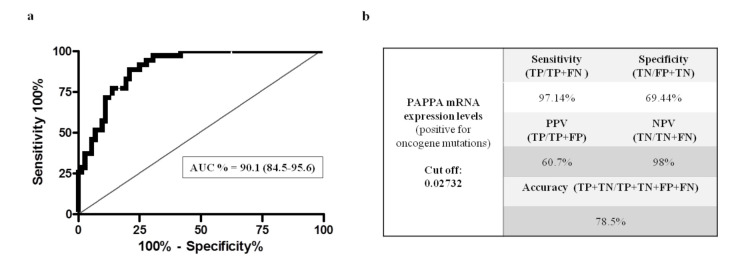
Diagnostic performance of the PAPPA mRNA expression levels in 107 cytological samples. (**a**) Receiver operating characteristic (ROC) curve and related area under the curve (AUC) of PAPPA expression to discriminate cytological samples positive for seven-gene panel. (**b**) The sensitivity, specificity, negative predictive value (NPV) and positive predictive value (PPV) of PAPPA expression with the cut off set at 0.02732. Abbreviations: TP, True positive; TN, true negative; FN, false negative; and FP, false positive.

**Figure 3 ijms-23-04648-f003:**
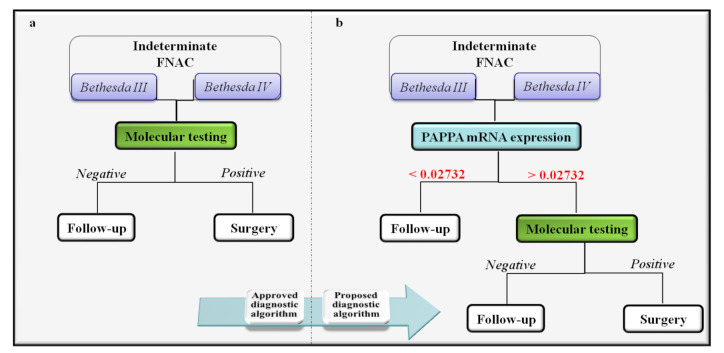
Classic management algorithm for indeterminate FNAC and the one proposed with the use of PAPPA mRNA expression as screening test. (**a**) Management algorithm for indeterminate thyroid nodules advocated by major societal guidelines. (**b**) Proposed management algorithm for indeterminate thyroid nodules introducing a PAPPA expression cut-off as new screening tool to select patients to be sent to molecular testing.

**Table 1 ijms-23-04648-t001:** Point mutations and rearrangements found in 35 cytological samples.

Gene	Cases (%)	Oncogenic Alteration	Cases (%)
*BRAF*	13/35 (37.1%)	BRAF V600E BRAF K601E	12/13 (92.3%) 1/13 (7.7%)
*RAS*		NRAS	9/17 (52.9%)
	17/35 (48.6%)	HRAS	7/17 (41.2%)
		KRAS	1/17 (5.9%)
*RET*	4/35 (11.4%)	RET/PTC1	4/4 (100%)
*PAX8*	1/35 (2.9%)	PAX8/PPARγ	1/1 (100%)

## Data Availability

Not applicable.

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
