# Peer review of "PAPPA Expression in Indeterminate Thyroid Nodules as Screening Test to Select Patients for Molecular Testing"

_ijms, 2022, doi:10.3390/ijms23094648_

Round 1

Reviewer 1 Report

I enjoyed reading this paper and I would commend the authors for their work. It is rare for me to read something and find no major flaws in it so I am happy to recommend its publication.

There are, perhaps,  a few minor grammatical changes:

Line 53, change “is” for “has”

Line 55, change “Due to this,” to “As a result.”

Line 58, change “evaluated” to “evaluate”

Line 59, change “sent to” to “sent for”

Line 126, change “part is” to “proportion are”

Personally, I would put the Materials and Methods section before the Results, rather than after the Discussion, but that’s only because I’m used to seeing them in that order.

The authors themselves indicate that the limitation in their work is that it should be confirmed with a larger study but, if that can be achieved, then their work could be a “game-changer” in this field.

Nicely done, everyone!

Author Response

We thank the Editor and the reviewers for their useful comments and suggestions that have been considered in the revised version. Changes are highlighted in yellow.

Reviewer #1:

Point 1: Few minor grammatical changes. Line 53, change “is” for “has”; Line 55, change “Due to this,” to “As a result.”; Line 58, change “evaluated” to “evaluate”; Line 59, change “sent to” to “sent for”; Line 126, change “part is” to “proportion are”.

Response 1: Thanks for your grammatical changes. All specific corrections have been incorporated into the text.

Point 2: Personally, I would put the Materials and Methods section before the Results, rather than after the Discussion, but that’s only because I’m used to seeing them in that order.

Response 2: Thanks for your suggestion. According to the "manuscript preparation" of the Journal, the Materials and Methods section should be reported after the Introduction, Results and Discussion sections.

Reviewer 2 Report

In this study, Marzocchi et al. found that Pregnancy-Associated Plasma Protein (PAPPA) expression increases in indeterminate thyroid nodules. They also showed that PAPPA expression has high sensitivity and negative predictive value to identify nodules with genetic alterations. The authors conclude that evaluating PAPPA expression in indeterminate thyroid cytology could represent a useful screening tool to select all patients that effectively need to be sent to molecular testing for better patient management. The data presented are interesting and the methodology employed appears correct. The paper requires minor changes.

Minor comments

  1. Lane 47: the authors should provide more information about the discovery of genetic alterations and which one is currently utilized.
  2. It would be better to separate figure 3. Figure 3a may become figure 1 in order to show and describe, in the introduction, the algorithm currently used. Figure 3b may become Figure 4.

Reviewer 3 Report

line 31: Early prenatal diagnosis. You should be more specific,  regarding prenatal diagnosis.  Choromosomal abnormalities, preeclampsia and intrauterine growth restriction risk may be added. 

At this point we need more recent references regarding the above combined risks.

How easy is to apply Papp-A measurement in FNA liquid, regarding the laboratories ability to provide such a measurement?

Can Papp-A be measured in the same kit,  as we do antenatally ? , (Roch etc).

If so, the paper is of high clinical importance.

Author Response

We thank the Editor and the reviewers for their useful comments and suggestions that have been considered in the revised version. Changes are highlighted in yellow.

Reviewer #2:

Point 1: line 31: Early prenatal diagnosis. You should be more specific, regarding prenatal diagnosis.  Choromosomal abnormalities, preeclampsia and intrauterine growth restriction risk may be added. At this point we need more recent references regarding the above combined risks.

Response 1: Following the Reviewer's comment, this information has been better specified into the text:  "...early prenatal diagnosis of choromosomal abnormalities, preeclampsia and intrauterine growth restriction risk" (lines 31,32). Furthermore, three more recent references have been added and numbered as 3,4 and 5.

Point 2: How easy is to apply Papp-A measurement in FNA liquid, regarding the laboratories ability to provide such a measurement?

Response 2: The assess of PAPPA mRNA expression in FNA liquid  is a simple, fast and cheap method for a molecular biology laboratory. The real time qPCR is a technique that is carried out in every molecular biology laboratory while more complex and expensive methods (such as Next Generation Sequence), that are used for oncogene mutations analysis, cannot be performed in all laboratories. For this reason, PAPPA mRNA expression in FNAC could represent a promising useful screening tool to be included in clinical practice.

Point 3: Can Papp-A be measured in the same kit,  as we do antenatally ? , (Roch etc)..

Response 3: Up to now, studies about the role of PAPPA in tumors have been conducted in tumor tissues or cultured cells and PAPPA expression has been evaluated with real time qPCR and Western blot. Differences between patients and controls have been detected mainly at tissue level and no in the circulation as in pregnancy, where PAPPA concentration increases steadily until term and is measurement in maternal serum with immunoassay (such as Roche etc..). Only a study detected PAPPA level with an ELISA kit in serum from pregnancy-associated breast cancer patients and healthy controls founding that PAPPA protein level was significantly increased in patients (Zhang J et al. Pregnancy-associated plasma protein-A promotes breast cancer progression. Bioengineered, 2022). However, the increase in PAPPA could be related to the pregnancy and not to the tumor. PAPPA in cancer does not appear to be released into the circulation at detectable levels different from healthy subjects (there are no studies). In this work, we have evaluated the mRNA expression level of PAPPA through a real time qPCR, a highly selective and sensitive technique. Anyway, it could be interesting to try to assay PAPPA in FNAC with an immunoassay to verify if it is measurable; although, it should be underlined that FNA sample from thyroid nodules often contains low concentration of material to examine with frequent contaminations.